# The impact of the 2014 Ebola epidemic on HIV disease burden and outcomes in Liberia West Africa

Soka J. Moses[1], Ian Wachekwa[2], Collin Van Ryn[3], Greg Grandits[3], Alice Pau[4], Moses Badio[1], Stephen B. Kennedy[1], Michael C. Sneller[4], Elizabeth S. Higgs[4], H. Clifford Lane[4], Mosoka Fallah[5], Stephen A. Migueles[4], Cavan Reilly[3]*

1 Partnership for Research on Vaccines and Infectious Diseases in Liberia (PREVAIL), Monrovia, Liberia, 2 John F Kennedy Medical Center, Monrovia, Liberia, 3 Division of Biostatistics, School of Public Health, University of Minnesota, Minneapolis, Minnesota, United States of America, 4 National Institutes of Health, Bethesda, Maryland, United States of America, 5 National Public Health Institute of Liberia, Monrovia, Liberia

☯ These authors contributed equally to this work.

* cavanr@biostat.umn.edu

**Data Availability Statement:** All relevant data are uploaded to Figshare and publicly accessible via the following DOI: 10.6084/m9.figshare.16441671.

## Abstract

### Background

Detailed longitudinal studies of HIV-positive individuals in West Africa are lacking. Here the HIV prevalence, incidence, all-cause mortality, and the proportion of individuals receiving treatment with cART in two cohorts of participants in Ebola-related studies are described.

### Setting

Individuals of all ages were enrolled and followed at four sites in the area of Monrovia, Liberia.

### Methods

Two cohorts identified in response to the Ebola epidemic are described to provide insights into the current state of the HIV epidemic. HIV testing was performed at baseline for participants in both cohorts and during follow-up in one cohort.

### Results

Prevalence and incidence of HIV (prevalence of 3.1% for women and 1.4% for men and incidence of 3.3 per 1,000) were higher in these cohorts compared to 2018 national estimates (prevalence of 1.3% and incidence of 0.39 per 1,000). Most participants testing positive did not know their status prior to testing. Of those who knew they were HIV positive, 7.9% reported being on antiretroviral treatment. The death rate among those with HIV was 12.3% compared to 1.9% in HIV-negative individuals (adjusted odds ratio of 6.87). While higher levels of d-dimer were associated with increased mortality, this was not specific to those with HIV, however lower hemoglobin levels were associated with increased mortality among those with HIV.

**Funding:** The project was funded in whole or in part with federal funds from the National Cancer Institute, National Institutes of Health, under Contract No. HHSN261200800001E. This research was supported [in part] by the National Institute of Allergy and Infectious Diseases.

**Competing interests:** The authors have declared that no competing interests exist.

## Conclusion

These findings point to a need to perform further research studies aimed at fulfilling these knowledge gaps and address current shortcomings in the provision of care for those living with HIV in Liberia.

## Introduction

Despite some progress, HIV infection continues to take a heavy toll in western and central Africa [1]. While annual new HIV infections declined by 13% and the number of AIDS-related deaths decreased by 29% between 2010 and 2018, those declines are lower than seen elsewhere. In eastern and southern Africa over this same period, the number of new infections declined by 28% and the number of AIDS-related deaths decreased by 44%. The most recent western and central African incidence-prevalence ratio of 5.5% [3.7–8.3%] is almost double the epidemic transition benchmark of 3%, the threshold below which the total population of people living with HIV will gradually fall. This metric also lags behind the value of 3.9% [3.0–5.1%] reported for eastern and southern Africa [1]. In the West African country of Liberia, there were an estimated 39,000 (36,000–44,000) people living with HIV in 2018; the HIV prevalence in the general population aged 15–49 years was 1.3% (1.1–1.4%) and the incidence was 0.39 (0.38–0.41) per 1,000 people [1]. HIV prevalence appears to be higher in urban settings (2.6%) compared to rural settings (0.8%), especially in the capital city Monrovia (3.2%) [2]. With an estimated incidence-prevalence ratio of 4.9% according to recent epidemic transition metrics, the HIV epidemic in Liberia remains an inadequately addressed public health problem [1].

HIV treatment in Liberia is provided vertically and free of charge by the Liberian Ministry of Health (MoH) through its National AIDS and Sexually Transmitted Infection Control Program (NACP). Even though efforts intensified within the last decade by the MoH to better organize services to treat and control the spread of HIV infection, numerous challenges have remained to combating the HIV epidemic in Liberia. These include lack of awareness of HIV status, low condom use, poverty, lack of education, and widespread stigma surrounding HIV/AIDS [3]. In addition, clinicians do not have regular access to measurements of CD4 counts and levels of HIV plasma viremia. Healthcare provision is also hampered by shortages of trained staff, especially in rural areas, erratic availability of HIV testing kits, and inconsistent access to combination antiretroviral therapy (cART) [3, 4]. The 2014–16 Ebola Virus Disease (EVD) outbreak in Liberia further strained the fragile healthcare system, including the NACP, which had already been weakened by years of civil strife. Thousands of patients living with HIV appear to have been lost to follow-up during and after the outbreak, with major declines estimated in access to HIV counselling, testing and treatment [5–7]. These factors may have led to increases in HIV transmission, resistance to cART, and accelerated disease progression [8, 9]. In 2016, only 19% of the people living with HIV in Liberia were accessing cART, and approximately 13% had suppressed viral loads [10]. In addition to the lack of services in some communities and low uptake of services even in places where they are available, data collection and health facility reporting are incomplete [11]. For these reasons, current in-depth information about the HIV epidemic in Liberia is lacking and significant gaps persist that minimize the likelihood of achieving the 95-95-95 targets that have been set for 2030.

Longitudinal studies of people living with HIV in western Africa are lacking. Such studies could provide valuable information about aspects of the HIV epidemic specific to this location. In this report, we investigated HIV/AIDS in participants of two large research studies

established after the beginning of the EVD outbreak in Liberia in 2014: a clinical trial of two Ebola vaccines and an EVD natural history study. Our aim was to describe HIV prevalence, incidence, all-cause mortality, and the proportion of individuals receiving treatment with cART in these cohorts in an effort to gain better insight into the HIV epidemic in Liberia.

## Methods

### Participants: PREVAIL cohorts

The two cohorts used in the analysis described here have been followed for up to four years (with some individuals in one cohort followed for as little as two years). Both cohorts were initiated under the auspices of the Partnership for Ebola Virus Research in Liberia (PREVAIL), an effort coordinated between the US Government and the MoH in response to the 2013–16 EVD outbreak in West Africa. All participants provided written informed consent or written informed consent was provided by parents or guardians, and written assents were obtained in accordance with international clinical research ethical norms. All protocols received approval from the NIAID Institutional Review Board and the National Research Ethics Board of Liberia.

PREVAIL I (ClinicalTrials.gov Identifier: NCT02344407) is a single center, double-blind, randomized, placebo-controlled study whose original objective was to determine the safety and efficacy of two experimental Ebola Virus (EBOV) vaccines compared to a common control. Vaccinations took place from February to April 2015. The details have been published elsewhere [12]. Due to the end of the outbreak, safety and immunogenicity objectives were substituted for the initial objective of efficacy. The enrollment, which was originally projected to exceed 28,000 participants was revised to 1,500. Only individuals with no history of EBOV infection and age 18 and over were enrolled and most participants were men. HIV and syphilis tests were conducted at baseline. Although the safety endpoint has been established, participants are still being followed under a modified protocol that seeks to assess long-term immunogenicity (participants consented to the modified protocol at the 1-year timepoint). To this end, participants have returned to the clinic on an annual basis and at these return visits, adverse events and survival are assessed and a blood draw is conducted for serology for EBOV (HIV tests were only conducted at baseline). Follow-up has been excellent with 98% of participants having been successfully followed for one year and with 1,251 (83%) returning for the year four visit.

PREVAIL III is a longitudinal cohort study whose primary objective was to determine the sequelae of EBOV infection. To this end, 1,144 EVD survivors and 2,784 controls were enrolled starting in June 2015. Enrollment took place at three sites that did not participate in PREVAIL I. The details of this study have also been described elsewhere [13]. To qualify as a survivor a participant needed to be listed on the MoH's listing of individuals recorded as positive for EBOV at an Ebola treatment unit during the outbreak. To qualify as a contact, one needed to be selected by a survivor as someone who had close contact with the survivor during the survivor's acute illness or had sexual contact after acute illness. Individuals of all ages were included, and the majority of participants were women. All participants return for a visit once every six months and at this visit, a blood draw for HIV and syphilis testing, a complete blood count with differential and common chemistries is conducted (blood draws were optional for participants less than 12 years of age). HIV and syphilis testing was done serum using the SD BIOLINE HIV/Syphilis Duo (06FK30, 06FK35) point-of-care diagnostic test (Standard Diagnostic, South Korea) [14]. The test is a solid phase immunochromatographic assay that qualitatively detects all antibody isotopes (IgG, IgM, IgA) specific to HIV 1/2 and/or *Treponema pallidum* (TP) simultaneously in human serum, plasma, or whole blood. The sensitivity and

specificity of the tests against HIV 1/2 and syphilis were previously found to be 100% in an African population of commercial specimens [14–16]. An extensive interview is conducted in which medication use (including cART) is determined, a detailed assessment of symptoms is collected, and a physical exam is conducted. In addition, deaths and hospitalizations are recorded based on self- or contact-reports. Follow-up has been good with 87.9% returning for the year two visit (the year three visit windows were still open for some participants when the analysis dataset was assembled).

According to the national guidelines, participants who tested HIV-positive were initially referred to an HIV clinic of their choice for treatment and care. All participants received HIV counselling at each study visit. Participants with a positive syphilis test or other clinically significant findings were referred to their primary care provider for treatment according to the national standard of care [17]. Both cohorts utilized the same laboratory resources and procedures and followed a common set of data management procedures. They also both relied on a group of community-based participant trackers who encouraged participants to return to the clinics for appointments. These commonalities provide for consistency across many aspects of the two cohorts, but participants were enrolled under different protocols designed for different objectives. Follow-up is still ongoing for both studies. The analysis presented here uses data up through April 2019.

## Statistical methods

Continuous measures were summarized by medians and quantiles while categorical variables were summarized with percentages. Tests for differences between individuals with and without HIV at baseline and between those who died or not were conducted using generalized mixed effects models which modeled correlations among measurements from related groups of survivors and contacts using an independence correlation structure and used generalized estimating equations to obtain parameter estimates [18]. Models for binary outcomes used logit link functions and those for continuous outcomes used linear link functions. Proportional hazards models [19] were used to test for covariates that may impact the incidence of HIV. Time to event analysis was conducted to estimate the impact of being HIV positive (at baseline or during follow-up) on survival and time until lost to follow-up. Kaplan Meier curves were computed, and log-rank tests were conducted. Cox proportional hazards were fit with covariate-time interaction terms [20] and Schoenfeld residuals [21] were examined to assess the proportionality assumption. All adjusted analyses controlled for age (treated as continuous), sex, site, and survivor status. A significance level of 0.05 was used to assess statistical significance and no correction for test multiplicity was used. Given the small number of hypothesis tests performed here corrections for test multiplicity were deemed unnecessary. There was no attempt to define certain data points as outliers.

Survival time was calculated as the number of months from study enrollment to date of death, or, if censored for death, date of the last study visit. PREVAIL III participants were considered lost to follow-up if they voluntarily discontinued follow-up or if their last study visit occurred more than 12 months prior to April 2019. PREVAIL I participants who did not consent to follow-up visits beyond 12 months were considered lost to follow-up if they missed the 12-month follow-up visit. PREVAIL I participants who consented to follow-up visits beyond 12 months were considered lost to follow-up if they missed both the 36 and 48-month follow-up visits. Time to lost-to-follow-up was calculated as the number of months from study enrollment to date of the last study visit plus nine months if lost to follow-up, or, if not lost to follow-up, date last seen. All statistical calculations were conducted using R version 3.2.

## Results

### HIV prevalence and incidence

The cohorts in PREVAIL I (vaccine recipients) and PREVAIL III (EVD survivors and contacts) had different gender compositions and HIV prevalence at baseline. At baseline in PREVAIL I, 78 (5.2%) of participants were HIV+ and 549 (36.6%) of participants were women (Table 1). The prevalence of HIV among women was 9.8% while the prevalence among men was 2.5%. At baseline, 76 (2.3%) of the participants enrolled on the PREVAIL III protocol were HIV+ and 1,836 (56.3%) of participants were women. The prevalence of HIV among women was 3.1% and the prevalence among men was 1.4%. We found an overall prevalence of 3.2% (Table 1). In PREVAIL I, 11.5% of those who were HIV positive at baseline self-reported as such while in PREVAIL III this was 6.6%.

Participants in PREVAIL III were retested for HIV every six months thereby allowing for computation of the incidence. Of the 3,187 participants with a median of 36 months of follow-up (quartiles 30 and 41), 29 new cases have been detected (Table 2), which implies an incidence of 3.3 cases per 1,000 person-years (95% CI: 2.2, 4.7 per 1,000 person-years).

### Death rates and loss to follow-up

Death rates were significantly higher among HIV+ individuals in both studies (Table 3; Fig 1). In PREVAIL I, the odds ratio for death associated with being HIV+ at baseline was 4.15 (95% CI, 1.87, 8.51), while in PREVAIL III, the odds ratio for death associated with HIV+ at baseline was 10.57 (95% CI, 4.08, 24.34). There were no deaths among the 29 participants identified as HIV positive during follow-up in PREVAIL III. Odds ratios adjusted for sex, age, survivor status and site also revealed significant associations between being HIV+ and death in both studies (PREVAIL I: adjusted OR = 4.87, 95% CI: 2.33, 10.17 and PREVAIL III: adjusted OR = 11.82, 95% CI: 4.93, 28.33). The association was highly significant in analyses that pooled data across studies (adjusted OR = 6.87, 95% CI: 3.78, 12.48).

In time to event analyses that examined death based on baseline HIV status, the hazard rate for death in PREVAIL I was 3.62 (95% CI: 1.85, 7.08) and 13.31 (95% CI: 5.71, 31.00) in PREVAIL III (Table 3). A hazard rate of 8.21 (95% CI: 4.38, 15.4) was found in pooled time to event analyses. Female gender and younger age were also identified to be protective from death (p<0.05).

One difference between the two studies was the substantial loss to follow-up in PREVAIL III associated with being diagnosed as HIV+ (Table 1; Fig 2). Among those who were HIV+, 13 (17.1%) were lost to follow-up compared to 302 (9.5%) among those who were HIV- (adjusted analyses: p = 0.03, OR = 1.98, 95% CI: 1.07, 3.64).

### Treatment and prognosis

Self-reported cART use was very low in both cohorts. In PREVAIL I, 9 (11.5%, 95% CI: 5.4%, 20.8%) HIV+ participants reported use of cART while in PREVAIL III, 6 (7.9%, 95% CI: 3.0%, 16.4%) reported use of cART at baseline. The proportion in PREVAIL III was significantly lower (p<0.01) than the UNAIDS reported value for Liberia of 19% (15–24%). Many of the participants in PREVAIL III initiated cART after being diagnosed, with 69.7% of participants reporting use during follow-up and 100% reporting continued use after first reported use.

In pooled analyses that adjusted for demographics, d-dimer levels were a factor of 1.5 higher among HIV+ individuals (p = 0.012, Table 4). Hemoglobin was about 10% lower among HIV+ individuals (p<0.001, Table 4). In addition, AST levels were significantly elevated among those with HIV (p = 0.015, Table 4) as was the prothrombin time (p = 0.002, only

**Table 1. Summary of characteristics, as well as number of deaths and number lost to follow-up, between HIV-positive and HIV-negative participants at PREVAIL I and PREVAIL III baseline.**

| | PREVAIL I | | | | PREVAIL III | | | | Combined cohorts | | | |
|---|---|---|---|---|---|---|---|---|---|---|---|---|
| | Overall | HIV- | HIV+ | *p*-value | Overall | HIV- | HIV+ | *p*-value | Overall | HIV- | HIV+ | *p*-value |
| Overall | 1500 | 1422 (94.8) | 78 (5.2) | | 3263 | 3187 (97.7) | 76 (2.3) | | 4763 | 4609 (96.8) | 154 (3.2) | |
| Female | 549 (36.6) | 495 (34.8) | 54 (69.2) | | 1836 (56.3) | 1780 (55.9) | 56 (73.7) | | 2385 (50.1) | 2275 (49.4) | 110 (71.4) | |
| Age in years at enrollment | 30 (24–38) | 29 (24–38) | 31 (26.2–39) | | 27 (18–38) | 28 (20–39) | 35 (28–41.2) | | 28 (20–38) | 29 (22–39) | 33.5 (27–41) | |
| On ART | 9 (0.6) | 0 (0) | 9 (11.5) | | 7 (0.2) | 1 (0) | 6 (7.9) | | 16 (0.3) | 1 (0) | 15 (9.7) | |
| Deaths | 65 (4.3) | 54 (3.8) | 11 (14.1) | < 0.001 | 43 (1.3) | 35 (1.1) | 8 (10.5) | < 0.001 | 108 (2.3) | 89 (1.9) | 19 (12.3) | < 0.001 |
| Lost to follow-up | 77 (5.1) | 71 (5) | 6 (7.7) | 0.404 | 315 (9.7) | 302 (9.5) | 13 (17.1) | 0.03 | 392 (8.2) | 373 (8.1) | 19 (12.3) | 0.02 |

Data are no. (%) or median value (interquartile range).

PREVAIL I models adjusted for sex and age. PREVAIL III and combined cohort models additionally adjusted for survivor status and enrollment site and used a generalized estimating equations approach to adjust for relationships between survivors and controls. PREVAIL I participants were categorized as controls in combined cohorts.

measured in PREVAIL III, Table 4). While these latter two differences were statistically significant, the magnitude of the differences was small and not suggestive of clinically significant differences that would be relevant for patient care. We did not detect significant differences in ALT (95% CI in HIV+ (9.4, 14.7) and HIV- (10.3, 12.3)) or eGFR (95% CI in HIV+ (106.2, 113.0) and HIV- (106.5, 108.5)).

Analyses that investigated the impact of d-dimer, hemoglobin and AST levels on all-cause mortality found significant effects for d-dimer (adjusted analyses: p = 0.004, OR = 1.14 for

**Table 2. Adjusted hazard ratios for HIV infection during PREVAIL III follow-up.**

| | PREVAIL III participants |
|---|---|
| | HIV-negative at baseline |
| | (n = 3187) |
| N (%) Contracted HIV | 29 (0.9) |
| Males | 12 (0.4) |
| Females | 17 (0.5) |
| Cases per 1000 person-years (95% CI) | 3.3 (2.2, 4.7) |
| Males | 3.1 (1.6, 5.4) |
| Females | 3.4 (2, 5.5) |
| Variable | |
| Indicator: female | 1.11 (0.53, 2.34) |
| Age in years at enrollment | 1.02 (0.99, 1.04) |
| Indicator: self-declared survivor | 0.87 (0.39, 1.97) |
| Enrollment site | |
| Site 2 | reference |
| Site 3 | 1.84 (0.72, 4.67) |
| Site 4 | 1.84 (0.73, 4.65) |

Data are hazard ratio (95% confidence interval) unless otherwise indicated.

Hazard ratios and confidence intervals were estimated by fitting a Cox proportional hazards model that included a random effect for relationships between survivors and contacts.

**Table 3. Adjusted hazard ratios and 95% confidence intervals for all-cause mortality by cohort.**

| Variable | PREVAIL I (n = 1500) HR (95% CI) | PREVAIL III (n = 3509) HR (95% CI) | Combined cohorts (n = 5009) HR (95% CI) |
|---|---|---|---|
| Indicator: HIV+ at baseline | 3.62 (1.85, 7.08) | 13.31 (5.71, 31) | 8.21 (4.38, 15.4) |
| Indicator: female | 0.76 (0.45, 1.28) | 0.57 (0.31, 1.07) | 0.65 (0.42, 1) |
| Age in years at enrollment | 1.06 (1.04, 1.07) | 1.06 (1.04, 1.08) | 1.06 (1.05, 1.08) |
| Indicator: self-declared survivor | | 1.08 (0.56, 2.07) | 1.08 (0.57, 2.07) |
| Enrollment site | | | |
| Site 1 | | | ref. |
| Site 2 | | ref. | 0.41 (0.22, 0.77) |
| Site 3 | | 0.66 (0.27, 1.61) | 0.27 (0.11, 0.64) |
| Site 4 | | 1.47 (0.72, 2.99) | 0.59 (0.3, 1.2) |

Abbreviations: HR, hazard ratio; CI, confidence interval; ref., reference level. Hazard ratios and confidence intervals were estimated by fitting Cox proportional hazards models. PREVAIL III models and combined cohort models included a random effect for relationships between survivors and contacts.

each standard deviation increase, 95% CI: 1.04, 1.24) and hemoglobin (adjusted analyses: p = 0.001, OR = 0.65 for each standard deviation increase, 95% CI: 0.5, 0.84) but failed to detect a significant effect for AST (adjusted analyses: p = 0.52, OR = 1.04 for each standard deviation increase, 95% CI: 0.92, 1.17). Analyses that investigated the potential for an interaction between HIV status and plasma biomarker levels failed to detect a significant interaction for d-dimer but did detect a significant interaction for hemoglobin (p = 0.02 for the interaction, OR = 0.77 among HIV negative participants, 95% CI: 0.57, 1.05 and OR = 0.28 among HIV positive participants, 95% CI: 0.13, 0.6). These results indicate that while d-dimer was predictive of mortality, this association was not restricted to those who were HIV positive, whereas the predictive utility of hemoglobin was restricted to HIV-positive individuals.

## Discussion

In this report, we investigated sub-populations from 5,428 individuals enrolled on two PRE-VAIL research protocols in Liberia. We found an overall prevalence of 3.2% and an annual incidence of 3.3 per thousand per year. Only 9.7% of those who were HIV positive were on

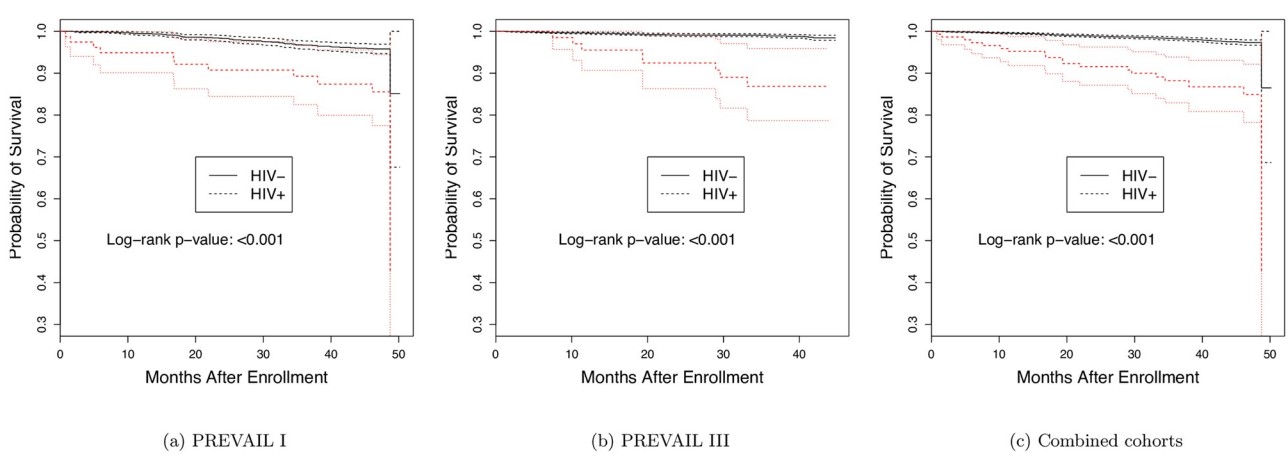

(a) PREVAIL I                    (b) PREVAIL III                    (c) Combined cohorts

**Fig 1. Probability of survival over time by baseline HIV status.**

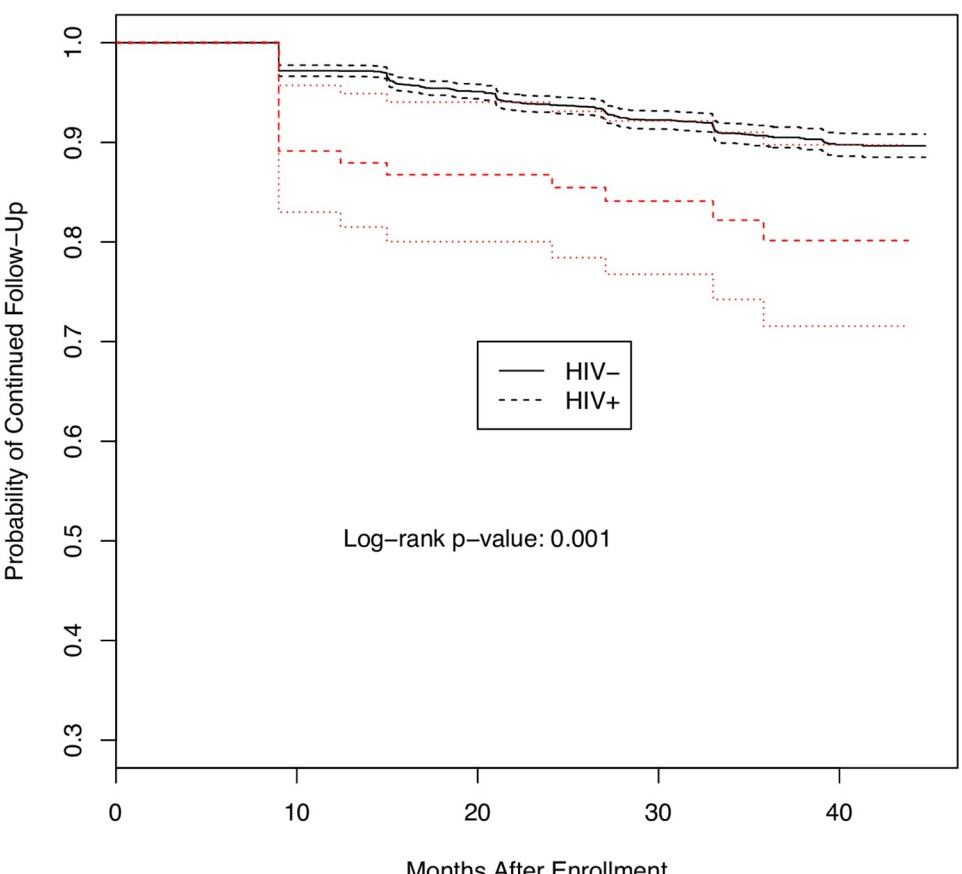

**Fig 2. Probability of loss to follow-up over time for participants in PREVAIL III by HIV status.**

**Table 4. Summary of baseline laboratories between HIV-positive and HIV-negative participants in PREVAIL I and PREVAIL III.**

| Laboratory | PREVAIL I | | | PREVAIL III | | | Combined cohorts | | |
|---|---|---|---|---|---|---|---|---|---|
| | HIV- | HIV+ | *p*-value | HIV- | HIV+ | *p*-value | HIV- | HIV+ | *p*-value |
| d-dimer ($\mu$g/mL) | 0.36 (0.27–0.57) | 0.56 (0.4–0.88) | 0.066 | 0.4 (0.3–0.7) | 0.7 (0.4–1.1) | 0.027 | 0.4 (0.3–0.6) | 0.6 (0.4–1) | 0.012 |
| eGFR (mL/min/ 1.73m$^2$) | 107.6 (91.9– 124.1) | 106.3 (91.5– 127.2) | 0.943 | 105.4 (89.3– 124.8) | 110.4 (98– 123.6) | 0.073 | 106 (90.1– 124.4) | 110.2 (95.3– 124.8) | 0.223 |
| ALT (U/L) | 6 (3–12) | 6 (2–11.8) | 0.473 | 6 (3–11) | 7 (3–12) | 0.222 | 6 (3–11) | 6 (2.2–12) | 0.538 |
| AST (U/L) | 13 (10–18) | 13 (10–19) | 0.999 | 13 (10–17) | 15 (11.8–21.2) | 0.009 | 13 (10–17) | 14 (11–20) | 0.015 |
| Hgb (g/dL) | 13.6 (12.5–14.6) | 12.1 (10.7–13.4) | < 0.001 | 13.2 (12.1–14.5) | 12.1 (10.7– 13.1) | < 0.001 | 13.4 (12.2– 14.6) | 12.1 (10.7–13.3) | < 0.001 |
| PT (sec) | . . . | . . . | | 13.9 (13.3–14.5) | 14.1 (13.6– 14.8) | 0.002 | . . . | . . . | |

Data are median value (interquartile range).

Abbreviations: eGFR, estimated glomerular filtration rate; ALT, alanine transaminase; AST, aspartate transaminase; Hgb, hemoglobin; PT, prothrombin time.

Table *p*-values are from generalized linear models that used an identity link function. PREVAIL I models adjusted for sex and age. PREVAIL III and combined cohort models additionally adjusted for survivor status and enrollment site and used a generalized estimating equations approach to adjust for relationships between survivors and controls.

PREVAIL I participants were categorized as controls in combined cohort models.

ART and overall individuals who were HIV positive were over six times more likely to die. Because participants enrolled in these two non-HIV research studies were not selected based on HIV risk factors, their risk of being HIV positive or acquiring HIV was unlikely to differ from that of the general population in the communities where they lived. However, we note that being HIV positive may increase the likelihood of EBOV infection due to immunosuppression and it may decrease the likelihood of surviving EVD. In fact the proportion of survivors who were HIV+ was lower than the proportion of contacts who were HIV+ but the difference failed to reach statistical significance [13]. This difference in rates may lead to a slight underestimate of prevalence overall. Due to ongoing health complications, which include reduced libido [13], the incidence of HIV among survivors might be less than the incidence in the overall population. We would not expect these potential differences in prevalence and incidence to impact the nature of the associations presented. Prevalence and incidence of HIV (prevalence of 3.1% for women and 1.4% for men and incidence of 3.3 per 1000) were higher in these cohorts compared to 2018 estimates (prevalence of 1.3% and incidence of 0.39 per 1000). We observed that if men and women had been represented equally in these two cohorts, the overall prevalence of HIV at baseline would have been 6.2% (95% CI: 4.8%, 7.5%) in PREVAIL I and 2.2% (95% CI: 1.7%, 2.7%) in PREVAIL III. Both are higher than previous national and county-level estimates [1, 22]. Similarly, based on proportional hazards models that examined the impact of sex, age at enrollment, self-declared survivor status, and enrollment site, the incidence of the disease over 36 months of follow-up was 3.3 cases per 1,000 person-years. This is more than eight times higher than the USAID and UNAIDS estimates for Liberia and the highest for any country in the West African sub-region compared to the UNAIDS 2018 estimates [23, 24].

We also observed that the hazard for death associated with being HIV+ was 8.2 in analyses that combined data across the cohorts. This likely reflected advanced HIV-associated immunodeficiency in untreated participants. This might have been indirectly potentiated by the impact of the EVD outbreak on the fragile healthcare system [5].

One difference between the PREVAIL I and PREVAIL III studies was the substantial preferential loss to follow-up of HIV-infected individuals in PREVAIL III. At the time of study entry, many of these individuals claimed to not know they were HIV+. Perhaps discovery of this and the subsequent emotional impact may have limited further participation. We note that this results in a negative bias in the difference of the death rates between HIV+ and HIV-, such that the odds ratio for death might be even larger than the numbers reported here.

Like Liberia, most countries in western and central Africa have a low prevalence of HIV ranging from 1–3.5% despite a relatively high incidence. This is likely a consequence of high mortality rates resulting partly from weak health systems due to inadequate investment in health [25]. High HIV/AIDS-associated morbidity and mortality is characteristic of a low prevalence epidemic [26]. Liberia is in the process of developing updated guidelines for prevention, testing, and treatment based on current global recommendations. The new treatment and care guidelines are widely expected to institutionalize the global test-and-start treatment approach and adapt current best practices in the care of HIV disease. Better access to treatment will lead to more people living with HIV/AIDS and a higher prevalence. Paradoxically, this will be a sign of improvement.

Consistent with other studies, we found that d-dimer levels were significantly elevated in HIV-positive patients compared with HIV-negative participants and were predictive of mortality. The odds ratio for mortality associated with a standard deviation increase in d-dimer levels was 1.14 (95% CI: 1.04, 1.24), however, this was not specific to HIV-infected individuals. Hemoglobin was lower in HIV+ participants and was also associated with mortality. The odds ratio associated with a standard deviation increase in hemoglobin levels was 0.65 (95% CI: 0.5,

0.84). Unlike d-dimer, while this association was significant in analyses that pooled HIV positive and negative individuals, analyses that did not pool these groups found the association was restricted to HIV positive individuals. Hemoglobin levels have been previously reported to be correlated with clinical progression of HIV and persistent anemia is an independent predictor of mortality [27–29]. While the observed differences for AST and prothrombin time were not clinically significant, elevated liver transaminases are frequently reported among HIV-infected individuals [30–33], however, these differences were not associated with differences in survival.

The main limitation of our study was that the two protocols were not designed a priori to study HIV/AIDS. Hence, the results should be interpreted with caution. In addition, little HIV-related information beyond what was described herein was collected, such as WHO clinical stage, CD4 counts, plasma HIV viral loads, resistance to antiretroviral drug regimens and the prevalence of HIV-related diseases. Finally, cART use, deaths and hospitalizations are self- or contact-reported.

Despite these limitations, to our knowledge, this is the largest longitudinal study with systematic collection of HIV status, treatment and mortality in research participants in West Africa. It creates opportunities to expand HIV research locally to help address some of the unanswered questions and provide a better understanding of the HIV epidemic in Liberia. More recently, a prospective observational cohort study of people living or newly diagnosed with HIV/AIDS was initiated in Liberia called PREVAIL VIII: A cohort clinical, viral, and immunological monitoring study of people living with retroviral infection in Liberia (HONOR). Data will be collected on a cohort of 3,000 people living with HIV to describe the social, demographic, clinical, immunologic, and virologic characteristics of HIV disease in two of the most affected counties in Liberia. A major benefit for participants of the study is immediate access to plasma viral load and CD4 count measurements and other diagnostic testing to inform clinical care. This kind of research is essential to focus action and investment and reposition the country on its track to achieve the 2030 global targets for HIV elimination as a disease of public health significance.

## Acknowledgments

The authors would like to thank the study participants. S.J.M. contributed to study conceptualization and data collection, I.W., A.P., S.B.K., E.S.H. and S.A.M. contributed to study conceptualization, C.V. and G.G. contributed to data analysis, M.B. contributed to data collection, M. C.S. and M.F. contributed to study design and data collection, H.C.L contributed to study design and conceptualization, and C.R. contributed to study conceptualization and data analysis. All coauthors assisted with drafting the paper and/or revision and interpretation of results. The content of this publication does not necessarily reflect the views or policies of the Department of Health and Human Services, nor does mention of trade names, commercial products, or organizations imply endorsement by the U.S. Government.

## Author Contributions

**Conceptualization:** Soka J. Moses, Ian Wachekwa, Alice Pau, Stephen B. Kennedy, Elizabeth S. Higgs, H. Clifford Lane, Mosoka Fallah, Stephen A. Migueles, Cavan Reilly.

**Formal analysis:** Collin Van Ryn, Greg Grandits, Cavan Reilly.

**Resources:** Soka J. Moses, Moses Badio, Michael C. Sneller, Mosoka Fallah.

**Writing – original draft:** Soka J. Moses, Ian Wachekwa, Stephen A. Migueles, Cavan Reilly.

**Writing – review & editing:** Soka J. Moses, Ian Wachekwa, Collin Van Ryn, Greg Grandits, Alice Pau, Moses Badio, Stephen B. Kennedy, Michael C. Sneller, Elizabeth S. Higgs, H. Clifford Lane, Mosoka Fallah, Cavan Reilly.

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
