## [Decision Letter · Decision Letter 0]

11 Jun 2021

PONE-D-21-03663

High mortality among HIV-infected individuals in a post-Ebola epidemic setting

PLOS ONE

Dear Dr. Reilly,

Thank you for submitting your manuscript to PLOS ONE. After careful consideration, we feel that it has merit but does not fully meet PLOS ONE’s publication criteria as it currently stands. Therefore, we invite you to submit a revised version of the manuscript that addresses the points raised during the review process.

Please take into account all the pertinent comments from the reviewers to improve the language, presentation of the data, and conclusions of the manuscript.

We look forward to receiving your revised manuscript.

Kind regards,

Graciela Andrei

Academic Editor

PLOS ONE

Journal Requirements:

3. Please ensure you have included the registration number for the clinical trial referenced in the manuscript.

4. Thank you for including your ethics statement:  "All protocols received approval from the NIAID Institutional Review Board and the National Research Ethics Board of Liberia.

All participants provided informed consent or consent was provided by caregivers and assents were obtained in accordance with international clinical research ethical norms.".  

Please provide additional details regarding participant consent. In the ethics statement in the Methods and online submission information, please ensure that you have specified what type you obtained (for instance, written or verbal, and if verbal, how it was documented and witnessed). If your study included minors, state whether you obtained consent from parents or guardians. If the need for consent was waived by the ethics committee, please include this information.

Reviewers' comments:

Reviewer's Responses to Questions

**Comments to the Author**

1. Is the manuscript technically sound, and do the data support the conclusions?

Reviewer #1: Yes

Reviewer #2: Yes

2. Has the statistical analysis been performed appropriately and rigorously? 

Reviewer #1: Yes

Reviewer #2: Yes

3. Have the authors made all data underlying the findings in their manuscript fully available?

Reviewer #1: Yes

Reviewer #2: Yes

4. Is the manuscript presented in an intelligible fashion and written in standard English?

Reviewer #1: Yes

Reviewer #2: Yes

5. Review Comments to the Author

Reviewer #1: The statistical methods appear to be high quality, though I've made a few suggestions to improve the language and reporting. The authors are a bit too zealous or cavalier in their conclusions. These are trial data and it's not normally a good assumption that people in a trial will be or behave the same as those outside a trial. I believe these are interesting analyses, but I think more caution should be taken regarding the conclusions and the discussion should be more measured.

1. (Statistical Methods section) Please provide citations for all statistical methods, preferably methodological over software.

2. (line 145) I'm curious why you chose an independence correlation structure over other structures that, potentially, would modeling the correlation between observations over time better.

3. (lines 143, 149, …) Is it the HIV status at baseline? In PREVAIL I, my understanding is HIV testing was only done at baseline. It might be a good idea to make this clear in this section.

4. (lines 151-2) I didn't understand what "covariate-time interaction terms" are. Time is part of the outcome, so I wasn't sure how interactions between covariates and time were included in the linear predictor. Do you mean time-dependent covariates?

5. (lines 294-6) I found this statement a little hard to accept. Maybe for PREVAIL I since the inclusion criteria are pretty minimal. Though, it's noted mostly men enrolled. For PREVAIL III, the sampling appears to be more specialized, so I'm not sure I am willing to go along with the statement that the pooling of these two cohorts is similar to the general population.

6. (line 345) The conclusions drawn and a good portion of the discussion hinge on the belief that the results are comparable to nationwide estimates and, for lack of a better term, "represent" the general population. It's confusing that now the authors are claiming the results from this study about HIV/AIDS should be interpreted with caution.

7. (Figures 1&2) Please include confidence bands on the survival estimates.

Reviewer #2: Your manuscript is well written, and the topic adressed is relevant, especially for resource-limited countries that very rarely conduct longitudinal studies to estimate HIV incidence. You just need to review a few things.

1. Abstract

The paragraph from line 23 to line 26 better justifies the results of this study. However, it is not very much in harmony with the title of the study. In addition, the elements mentioned in this paragraph do not appear very clearly in the general introduction of the body of this manuscrit. As the abstract is the synthesis of the body of the manuscrit, in my opinion, there should be no element in the abstract that isn't in the body of the article. I suggest that you harmonise the justification that appears in the abstract of the study with that which appears in the general introduction to the body of the manuscrit.

2. Introduction

- You have provided enough information on the weaknesses in access to HIV health care services; however, there is little information on the overall organisation of HIV health care services in Liberia. It would be important for you to briefly describe the current organisation of HIV health care provision in Liberia. This would better support the weaknesses that you were willing to address in your manuscript.

- From Line 72 to Line 73 you have mentionned that "Thousands of patients living with HIV were lost to follow-up during and after the Ebola outbreak, with significant declines reported in access to HIV counselling, testing and treatments", but there are not enough statistics to support the claims about the decline in the use of health services during and after the Ebola epidemic. In addition, it would be important for you to report on the proportion of the lost to follow-up patients among the persons which are living with HIV in treatment centres in Liberia.

- The justification for your study that you mention in lines 77 to 81 seems to me to be a bit global, not too specific. It isn't very specific to the title of your manuscript. I suggest you improve this part.

- The objectives of your study go beyond mortality (you talk about prevalence, incidence and mortality). However, the title of your study focuses only on mortality. In my opinion, there is a need to adapt the title of your study

3. Methods

- In lines 124 to 125 you indicate that blood tests were taken from participants who returned for a clinical visit. This was done to test for HIV and Syphilis. In my opinion, it is important to clarify the nature of the tests used and the algorithm for HIV testing and syphilis testing. It is also important to explain a bit more how the management of HIV or syphilis positive cases was carried out.

- In line 116, you mention "1,144 Ebola survivors and 2,784 controls" without explaining the criteria for matching controls to cases. I suggest that you provide more details.

- Were there any lost to follow-up cases who returned to the cohort? If so, I suggest you report this in the methodology; and say how the data for these cases was managed?

4. Results

No comment

5. Discussion

The result: "We found an overall prevalence of 3.2% and an annual incidence of 3.3 per thousand per year" mentioned here, in lines 191 and 192, is not very explicit in the result chapter. In my opinion, these statistics should be mentioned in the result chapter in lines 171 to 179.

6. PLOS authors have the option to publish the peer review history of their article (what does this mean?). If published, this will include your full peer review and any attached files.

Reviewer #1: No

Reviewer #2: **Yes: **Niouma Nestor LENO

---

## [Author Response · Author response to Decision Letter 0]

16 Jul 2021

Response to Reviewers

PONE-D-21-03663

High mortality among HIV-infected individuals in a post-Ebola epidemic setting

PLOS ONE

Journal Requirements: When submitting your revision, we need you to address these additional requirements.

https://journals.plos.org/plosone/s/file?id=ba62/PLOSOne_formatting_sample_title_authors_affiliations.pdf [journals.plos.org]

We have checked our manuscript for conformance to these style requirements and believe we are compliant.

We have revised the reference list to include citations for the statistical methods at the request of one of the reviewers.

3. Please ensure you have included the registration number for the clinical trial referenced in the manuscript.

We now include the clinical trials.gov identifier in the manuscript (page 5 line 104).

4. Thank you for including your ethics statement: "All protocols received approval from the NIAID Institutional Review Board and the National Research Ethics Board of Liberia. All participants provided informed consent or consent was provided by caregivers and assents were obtained in accordance with international clinical research ethical norms.".

Please provide additional details regarding participant consent. In the ethics statement in the Methods and online submission information, please ensure that you have specified what type you obtained (for instance, written or verbal, and if verbal, how it was documented and witnessed). If your study included minors, state whether you obtained consent from parents or guardians. If the need for consent was waived by the ethics committee, please include this information.

We have modified our description of the consent process to describe the type: “All participants provided written informed consent or written informed consent was provided by parents or guardians and written assents were obtained in accordance with international clinical research ethical norms.” This is on page 5 lines 99-102.

For additional information about PLOS ONE ethical requirements for human subjects research, please refer to

http://journals.plos.org/plosone/s/submission-guidelines#loc-human-subjects-research [journals.plos.org].

5. We note that the grant information you provided in the “Funding Information” and “Financial Disclosure” sections do not match. When you resubmit, please ensure that you provide the correct grant numbers for the awards you received for your study in the “Funding Information” section.

We have addressed this in the resubmission.

Reviewer #1: The statistical methods appear to be high quality, though I've made a few suggestions to improve the language and reporting. The authors are a bit too zealous or cavalier in their conclusions. These are trial data and it's not normally a good assumption that people in a trial will be or behave the same as those outside a trial. I believe these are interesting analyses, but I think more caution should be taken regarding the conclusions and the discussion should be more measured.

1. (Statistical Methods section) Please provide citations for all statistical methods, preferably methodological over software.

Thank you for reviewing our manuscript. As suggested, we have added 4 references to the statistical methods section (page 8).

2. (line 145) I'm curious why you chose an independence correlation structure over other structures that, potentially, would modeling the correlation between observations over time better.

We used an independence correlation structure because it is relatively robust to the true correlation structure. While there may be some loss of efficiency, there is less of a danger of a major loss of efficiency compared to using an incorrect correlation structure that deviates from independence. This is demonstrated in the Liang and Zeger paper now referenced in this section. 

3. (lines 143, 149,) Is it the HIV status at baseline? In PREVAIL I, my understanding is HIV testing was only done at baseline. It might be a good idea to make this clear in this section.

We agree and have modified the text to be more explicit about baseline and incident HIV status (page 7-8 line 159-160).

4. (lines 151-2) I didn't understand what "covariate-time interaction terms" are. Time is part of the outcome, so I wasn't sure how interactions between covariates and time were included in the linear predictor. Do you mean time-dependent covariates?

Thank you for making this point. The unspecified baseline hazard function in the Cox model is potentially (and generally assumed to be) dependent on time. By including terms that depend on time and the covariate, one can evaluate if the impact of the covariate acts in a manner that is consistent with the proportional hazards assumption. We have provided a reference (i.e., Hess) that describes how to do this using cubic splines (page 8 line 170). 

5. (lines 294-6) I found this statement a little hard to accept. Maybe for PREVAIL I since the inclusion criteria are pretty minimal. Though, it's noted mostly men enrolled. For PREVAIL III, the sampling appears to be more specialized, so I'm not sure I am willing to go along with the statement that the pooling of these two cohorts is similar to the general population.

You raise an important point. We agree that the broad inclusion criteria of PREVAIL I make it likely that those study participants were likely similar to the community from which they came, with the exception that men were over-represented. For this reason, we have been careful to stratify or control for gender in the summaries we compute. The entry criteria for PREVAIL III were either being an Ebola survivor or having a contact who is an Ebola survivor. The contacts were drawn from the same communities, and mostly from the same families, as Ebola survivors so the survivors and contacts should be similar to one another with regard to many features, including HIV risk factors. Thus, the similarity of PREVAIL III participants to the larger community largely comes down to the similarity of Ebola survivors to the larger population. There are likely individual characteristics that make it more or less likely that one will contract and survive an Ebola infection. These are likely related to occupation and overall health status and potentially includes HIV infection (and so HIV risk factors). This would make the population of PREVAIL III participants less likely to be HIV positive, so there is some potential for underestimation of prevalence and incidence of HIV. However, we don’t think this would alter the nature of the association between HIV status and other factors explored (e.g., d-dimer). We have altered the discussion to make these points (pages 17-18 lines 330-338). 

6. (line 345) The conclusions drawn and a good portion of the discussion hinge on the belief that the results are comparable to nationwide estimates and, for lack of a better term, "represent" the general population. It's confusing that now the authors are claiming the results from this study about HIV/AIDS should be interpreted with caution.

We apologize for the confusion. The advice to interpret the results with caution at the referenced point in the manuscript is based on the fact that this is a secondary analysis of data generated to address different sets of questions than those investigated here. It is not based on concerns about the ability of these data to represent the general population. As noted in this reviewer’s previous comments and our response, there is some concern regarding the “representativeness” of these cohorts and as noted we now note this and provide some supporting arguments for this point.

7. (Figures 1&2) Please include confidence bands on the survival estimates.

This is a good suggestion and we have included these in the resubmission.

Reviewer #2: Your manuscript is well written, and the topic adressed is relevant, especially for resource-limited countries that very rarely conduct longitudinal studies to estimate HIV incidence. You just need to review a few things.

1. Abstract

The paragraph from line 23 to line 26 better justifies the results of this study. However, it is not very much in harmony with the title of the study. In addition, the elements mentioned in this paragraph do not appear very clearly in the general introduction of the body of this manuscrit. As the abstract is the synthesis of the body of the manuscrit, in my opinion, there should be no element in the abstract that isn't in the body of the article. I suggest that you harmonise the justification that appears in the abstract of the study with that which appears in the general introduction to the body of the manuscrit.

Thank you for reviewing our manuscript and making this valuable point. We have expanded the introduction to ensure these points are incorporated (page 4-5 lines 85-92).

2. Introduction

- You have provided enough information on the weaknesses in access to HIV health care services; however, there is little information on the overall organisation of HIV health care services in Liberia. It would be important for you to briefly describe the current organisation of HIV health care provision in Liberia. This would better support the weaknesses that you were willing to address in your manuscript.

Thank you for this comment. Although a description of the overall organization of HIV care services in Liberia would be quite challenging to summarize briefly, we have made some revisions to the Introduction that now includes efforts made by the Liberian government to better organize services for people living with HIV infection (page 3 lines 63-67).

- From Line 72 to Line 73 you have mentionned that "Thousands of patients living with HIV were lost to follow-up during and after the Ebola outbreak, with significant declines reported in access to HIV counselling, testing and treatments", but there are not enough statistics to support the claims about the decline in the use of health services during and after the Ebola epidemic. In addition, it would be important for you to report on the proportion of the lost to follow-up patients among the persons which are living with HIV in treatment centres in Liberia.

The reviewer makes a fair point here. We have scaled this back some in the Introduction to account for the lack of robust numbers. As such, reporting the proportion of persons living with HIV who were lost to follow up in treatment centers in Liberia is not something that can be reliably done beyond what is presented in the cited references. These changes are on page 4 lines 75-78.

- The justification for your study that you mention in lines 77 to 81 seems to me to be a bit global, not too specific. It isn't very specific to the title of your manuscript. I suggest you improve this part.

Our goal was to summarize the status of persons living with HIV in Liberia during and shortly after the 2014 Ebola epidemic with data obtained from 2 studies of Ebola. In the portion of the manuscript referenced here, we note that this was motivated by a lack of HIV services and accurate reporting resulting from the breakdown in the health care system due to the Ebola epidemic. We have altered the title to better reflect the results presented in this manuscript. We have also updated the 90-90-90 goal to the current 2030 95-95-95 goal.

- The objectives of your study go beyond mortality (you talk about prevalence, incidence and mortality). However, the title of your study focuses only on mortality. In my opinion, there is a need to adapt the title of your study

This is a good point and we have modified the title to better reflect the content of the manuscript. Thanks for this suggestion.

3. Methods

- In lines 124 to 125 you indicate that blood tests were taken from participants who returned for a clinical visit. This was done to test for HIV and Syphilis. In my opinion, it is important to clarify the nature of the tests used and the algorithm for HIV testing and syphilis testing. It is also important to explain a bit more how the management of HIV or syphilis positive cases was carried out.

We agree and we now provide this information. The additional information has been presented on page 6-7 lines 133-139 and page 7 lines 145-149.

- In line 116, you mention "1,144 Ebola survivors and 2,784 controls" without explaining the criteria for matching controls to cases. I suggest that you provide more details.

We provided a description of how cases and controls were matched: “To qualify as a contact, one needed to be selected by a survivor as someone who had close contact with the survivor during the survivor’s acute illness or had sexual contact after acute illness.” This is now on page 6 lines 127-129.

- Were there any lost to follow-up cases who returned to the cohort? If so, I suggest you report this in the methodology; and say how the data for these cases was managed?

Some participants missed visits, but they were not withdrawn from the study due to missed visits.

4. Results

No comment

5. Discussion

The result: "We found an overall prevalence of 3.2% and an annual incidence of 3.3 per thousand per year" mentioned here, in lines 191 and 192, is not very explicit in the result chapter. In my opinion, these statistics should be mentioned in the result chapter in lines 171 to 179.

We now include the prevalence estimate in the text at the location suggested by the reviewer. The incidence estimate is given on page 11 lines 216-218 in the results section, not just in the discussion.

---

## [Decision Letter · Decision Letter 1]

23 Aug 2021

The impact of the 2014 Ebola epidemic on HIV disease burden and outcomes in Liberia West Africa

PONE-D-21-03663R1

Dear Dr. Reilly,

We’re pleased to inform you that your manuscript has been judged scientifically suitable for publication and will be formally accepted for publication once it meets all outstanding technical requirements.

Kind regards,

Graciela Andrei

Academic Editor

PLOS ONE

Additional Editor Comments (optional):

Reviewers' comments:

Reviewer's Responses to Questions

**Comments to the Author**

1. If the authors have adequately addressed your comments raised in a previous round of review and you feel that this manuscript is now acceptable for publication, you may indicate that here to bypass the “Comments to the Author” section, enter your conflict of interest statement in the “Confidential to Editor” section, and submit your "Accept" recommendation.

Reviewer #1: All comments have been addressed

Reviewer #2: All comments have been addressed

2. Is the manuscript technically sound, and do the data support the conclusions?

Reviewer #1: (No Response)

Reviewer #2: Yes

3. Has the statistical analysis been performed appropriately and rigorously? 

Reviewer #1: (No Response)

Reviewer #2: Yes

4. Have the authors made all data underlying the findings in their manuscript fully available?

Reviewer #1: (No Response)

Reviewer #2: Yes

5. Is the manuscript presented in an intelligible fashion and written in standard English?

Reviewer #1: (No Response)

Reviewer #2: Yes

6. Review Comments to the Author

Reviewer #1: (No Response)

Reviewer #2: After reading the entire new version of your manuscript, I find that it has taken into account almost all of my previous comments.

From my side, it's okay for publication.

Thank you

7. PLOS authors have the option to publish the peer review history of their article (what does this mean?). If published, this will include your full peer review and any attached files.

Reviewer #1: No

Reviewer #2: **Yes: **Niouma Nestor LENO

---

## [Editor Report · Acceptance letter]

2 Sep 2021

PONE-D-21-03663R1 

The impact of the 2014 Ebola epidemic on HIV disease burden and outcomes in Liberia West Africa 

Dear Dr. Reilly:

I'm pleased to inform you that your manuscript has been deemed suitable for publication in PLOS ONE. Congratulations! Your manuscript is now with our production department. 

Kind regards, 

on behalf of

Dr. Graciela Andrei 

Academic Editor

PLOS ONE